# The Tango between Cancer-Associated Fibroblasts (CAFs) and Immune Cells in Affecting Immunotherapy Efficacy in Pancreatic Cancer

**DOI:** 10.3390/ijms24108707

**Published:** 2023-05-13

**Authors:** Imke Stouten, Nadine van Montfoort, Lukas J. A. C. Hawinkels

**Affiliations:** Department of Gastroenterology and Hepatology, Leiden University Medical Center, 2333 ZA Leiden, The Netherlands; i.stouten@lumc.nl

**Keywords:** tumour microenvironment, desmoplasia, cancer-associated fibroblasts, immune cells, T cells, pancreatic ductal adenocarcinoma, immunotherapy, checkpoint inhibitor

## Abstract

The lack of response to therapy in pancreatic ductal adenocarcinoma (PDAC) patients has contributed to PDAC having one of the lowest survival rates of all cancer types. The poor survival of PDAC patients urges the exploration of novel treatment strategies. Immunotherapy has shown promising results in several other cancer types, but it is still ineffective in PDAC. What sets PDAC apart from other cancer types is its tumour microenvironment (TME) with desmoplasia and low immune infiltration and activity. The most abundant cell type in the TME, cancer-associated fibroblasts (CAFs), could be instrumental in why low immunotherapy responses are observed. CAF heterogeneity and interactions with components of the TME is an emerging field of research, where many paths are to be explored. Understanding CAF–immune cell interactions in the TME might pave the way to optimize immunotherapy efficacy for PDAC and related cancers with stromal abundance. In this review, we discuss recent discoveries on the functions and interactions of CAFs and how targeting CAFs might improve immunotherapy.

## 1. Introduction

Pancreatic ductal adenocarcinoma (PDAC) is the most common type of pancreatic cancer, accounting for more than 90% of pancreatic cancer cases [1]. Of all cancer types, pancreatic cancer has the lowest 5-year survival of only 8% [2]. Treatment for pancreatic cancer consists of a combination of surgery and systemic therapies. Chemotherapy is the most common therapy for PDAC since 80–90% of the patients are precluded from surgical resection due to the advanced state of their disease [3,4]. Nonetheless, chemoresistance is a recurrent problem [5]. Recently, immunotherapy as an alternative approach to target the tumour has become of interest. The goal of immunotherapy is (re-)activating the immune system directed against cancer. Multiple novel immunotherapies have been studied in PDAC, which often aim to activate T cells either by blocking their inhibitory signals or by enhancing their anti-tumour activity. Among these therapies are immune checkpoint inhibitors, cancer vaccines, CD40 agonists, chimeric antigen receptor (CAR) T cells and oncolytic viruses [6,7].

Immunotherapies have yet to reach a clinical effect in PDAC. The low mutational burden in PDAC is one of the potential explanations underlying the low response rates [8]. In addition, the nature of the tumour microenvironment (TME) of PDAC is believed to play a decisive role in response to various therapies. This TME is characterized by desmoplasia (enhanced extracellular matrix deposition), poor vascularization, low oxygen levels and low immune cell infiltration [9]. The TME is composed of a variety of cells, including cancer-associated fibroblasts (CAFs), endothelial cells and immune cells, accompanied by a few other cell types with lower abundance [10]. The most abundant cell type, CAFs, can be subdivided into different subsets [11]. However, the classification of different CAF subtypes in PDAC is still in its infancy. An increasing number of studies suggest that CAFs could play a part in why immunotherapies have not reached their full potential [12]. The presence of CAFs could for example contribute to exclusion of T cells from the tumour, often referred to as an excluded or cold tumour. Since immunotherapy relies on the presence of its effector cells, the T cells, T cell exclusion in the TME impairs efficacy [13]. Hence, studying CAF heterogeneity, their functions and how to therapeutically target them could provide insight into immunotherapy resistance and provide directions to improve immunotherapy efficacy. In this review, we summarize the current knowledge of how the interactions of CAFs in the TME affect immunotherapy responses in PDAC.

## 2. CAFs

A consensus statement defined CAFs as cells with an elongated morphology that lack cancer-specific mutations and are negative for epithelial, leukocyte or endothelial markers [14]. CAFs are thought to predominantly originate from local resident fibroblasts with a contribution from several other cell types [15]. In a cancerous environment, fibroblasts can become stimulated by high local concentrations of cytokines and adopt a different, activated phenotype [16]. Solid tumours can consist, for a large part, of CAFs, representing up to 80% of the PDAC and breast cancer tumour mass [17]. Studies have shown that CAFs in different cancer types can have different molecular and functional characteristics. For instance, CAFs in PDAC have been found to express different markers and secrete other factors than CAFs in breast cancer, colorectal cancer and lung cancer [18]. The pro-tumourigenic properties of CAFs can be exerted through extracellular matrix (ECM) remodelling, direct cell–cell interactions, as well as secretion of soluble factors [19]. In PDAC, effects of CAFs on tumour cells include increased proliferation, migration and invasion, for instance via indirect interleukin (IL)-6 signalling [20,21]. These interactions between CAFs and tumour cells contribute to the formation of distant metastasis. Besides tumour cells, CAFs interact with endothelial cells through the upregulation of vascular endothelial growth factor (VEGF) [22]. Hypoxia stimulates CAFs to induce the proliferation and migration of endothelial cells, resulting in angiogenesis in vitro and in vivo [23]. CAFs importantly could impact the immune status of the tumour as well [13,24]. Thus far, the role of CAFs in immune deserted (cold) tumours is quite unknown. As for inflamed (hot) tumours, the secretory profile of CAFs could influence immune activity via inhibition of cytotoxic functions or recruitment of immunosuppressive cells [13]. CAFs have been correlated with T cell exclusion in solid tumours, which suggests CAFs being linked to an immune excluded phenotype [24]. Activation of CAFs by transforming growth factor β (TGF-β) promotes mediation of T cell exclusion and has been correlated to checkpoint inhibitor resistance [13,24]. Furthermore, CAFs increase levels of fibronectin and collagen that alter the composition of the ECM, which promotes the migratory properties of tumour cells [25]. Conceivably, the relationship between CAFs and other cells in the TME plays a key role in tumour progression.

### 2.1. The Good or the Bad Guys?

In PDAC, as in many other cancer types, data on the role of CAFs in tumour progression pointed towards the pro-tumourigenic effects of CAFs [26,27,28]. In vitro, pancreatic cancer cells displayed enhanced cell proliferation and migration in the presence of CAFs [26,28]. Moreover, in vivo depletion of fibroblast activation protein (FAP)+ stromal cells, a marker abundantly expressed by CAFs, resulted in the inhibition of tumour growth [27]. Recently, the presence of a CAF-based gene expression signature created with PDAC expression data from the Cancer Genome Atlas (TCGA), Gene Expression Omnibus (GEO) and ArrayExpress was related to worse patient survival rates [29]. Even though the vast majority of published studies exclusively reported pro-tumourigenic properties of CAFs in PDAC, more recent work showed opposite findings. Transgenic mice with α-smooth muscle actin (α-SMA)+ driven CAF depletion developed undifferentiated, aggressive tumours. The overall survival (OS) of the mice was decreased, and the number of regulatory T cells in the tumour was significantly enhanced [30]. In another study, inhibition of sonic hedgehog (shh) signalling in mice with PDAC resulted in a decrease in α-SMA+ CAFs in the TME, accompanied by the presence of aggressive and undifferentiated tumours [31]. These data suggest that α-SMA+ CAFs might pose anti-tumourigenic effects in PDAC and other cancer models.

The discovery that the presence/depletion of different CAF subsets can lead to opposite outcomes reveals the heterogeneity of the cell population. CAFs in PDAC are most often divided into three subtypes: myofibroblast-type CAFs (myCAFs), inflammatory CAFs (iCAFs) and antigen-presenting CAFs (apCAFs) [32,33]. Defining these subsets could improve our understanding of their specific effects on the TME and their role in affecting immunotherapy resistance.

### 2.2. MyCAFs, iCAFs and apCAFs

iCAFs and myCAFs were the first subsets of CAFs recognized as two distinct types in PDAC. ECM-producing α-SMA+ CAFs located close to epithelial tumour cells in human PDAC tissue and mouse models for pancreatic cancer were defined as myCAFs [32]. The most used model for PDAC is the KPC mouse. These mice spontaneously develop pancreatic cancer lesions due to pancreas-specific point mutations in protein 53 (P53) and an activating mutation in Kirsten rat sarcoma virus (KRAS) to recapitulate human pancreatic cancer [34]. The research was continued with a co-culture of pancreatic stellate cells, a possible precursor of CAFs, and PDAC organoids [32]. Pancreatic stellate cells adjacent to the organoid became myCAFs, and a noticeable increase in inflammatory cytokines such as interleukin 6 (IL-6) by another CAF subset was observed. The cytokine-secreting CAFs did not express α-SMA. This led to the revelation of two distinct subtypes: myCAFs with high α-SMA expression and iCAFs with low α-SMA expression and elevated cytokine expression. By RNA sequencing, the division of the iCAFs and myCAFs was confirmed since the two subclasses had unique clusters of upregulated genes [32]. Later on, the signalling pathways of the two different phenotypes were studied. Induction of iCAFs is promoted by IL-1 and involves JAK/STAT signalling, whereas TGF-β counteracts this cascade by downregulating IL-1 receptor expression, which leads to differentiation into myCAFs. The two subclasses exhibit a degree of plasticity since it has been shown that iCAFs switch to a myCAF phenotype once IL-1 signalling is inhibited. Along with these results, a small CAF population with both α-SMA and activated STAT signalling was identified, which also supports the plasticity between the two subcategories [35]. More recently, a third subset has been added. A subpopulation of CAFs that express major histocompatibility complex class II (MHC II) was able to activate CD4 T cells, which gave rise to their name, antigen-presenting CAFs (apCAFs) [33]. To conclude, to date, three well-described subtypes of CAFs can be distinguished that are associated with different functions in the TME (Figure 1). However, these phenotypes are plastic, and their functions intertwine with each other.

### 2.3. The Different Colours beyond Black and White

As research evolved, it became more apparent that the division into three subtypes of CAFs is probably somewhat too simplistic. In other cancer types, such as melanoma, lung cancer and breast cancer, more CAF subsets have been reported [36,37]. In PDAC, recent work, for example, identified CAF heterogeneity by the presence or absence of endoglin (CD105), a glycoprotein that is part of the TGF-β receptor complex [30]. Notably, CD105− CAFs included myCAFs, iCAFs and apCAFs, while CD105+ CAFs included only myCAFs and iCAFs. CD105+ CAFs are the largest in number and exhibit tumour-permissive properties in a pancreatic cancer mouse model. CD105− CAFs appear more tumour suppressive through activation of helper T cells and cytotoxic T cells. In contrast, evidence from multiple studies supports that apCAFs, which are observed in the CD105− group, cause immunosuppression by inhibiting T cells and are therefore considered pro-tumourigenic [33,38,39,40]. However, the functional role of CD105+ and CD105− subsets is still not firmly established, as data from our group show that therapeutic inhibition or fibroblast-specific genetic deletion of CD105 did not affect mouse pancreatic tumour growth [41,42].

Another study has endeavoured on subcategorizing CAFs based on their metabolic state. In PDAC patients with low desmoplasia, CAFs with a highly active metabolic state (meCAFs) were increased in comparison to patients with highly desmoplastic tumours [43]. The risk of metastasis in patients with an abundant presence of meCAFs was increased, yet their response to immunotherapy was profoundly better. Recently, a follow-up study from the same group identified phospholipase A2 group 2A (PLA2G2A) as a prominent marker for meCAFs [44]. Besides research on CAFs in the primary tumour, CAFs in metastases have been studied and are referred to as a separate subclass, metastases-associated fibroblasts (MAFs), due to observed differences in function. The interactions between MAFs and tumour cells mainly support angiogenesis and tumour progression [45]. Overall, the deviations from the status quo of CAF subclassifications show that the definition of existing subclasses is not definite. These subclasses are probably intertwined, and specific subsets could be more causally involved in the low response to immunotherapies than others.

## 3. CAF-Immune Cell Interactions

Immunotherapy is aimed at inducing or exploiting immune cell reactivity towards malignant cells. In PDAC, the immunosuppressive TME limits the efficacy of immunotherapeutic approaches. Both immune cells of the myeloid and lymphoid lineage take part in creating this immunosuppressive environment, and accumulating evidence suggests that interactions with CAFs can play a determining role [46,47]. Myeloid cells are a component of the innate immune system that rapidly infiltrate local tissue sites upon their recruitment. Myeloid cells that are associated with PDAC progression are myeloid-derived suppressor cells (MDSCs), tumour-associated macrophages (TAMs) and tumour-associated neutrophils (TANs) [46]. Within the lymphoid lineage, T cells have been studied extensively in PDAC. Cytotoxic T cells are direct anti-cancer effector cells, whereas helper T cells act indirectly by activating other immune cells, including cytotoxic T cells. Regulatory T cells, in contrast, suppress T cell activity. The balance between these T cell subsets plays a crucial role in anti-tumour immunity and therefore tumour progression, and their activity heavily depends on the other cells present in the stroma [48,49]. Reported mechanisms on how CAF–immune cell interactions can shape the immunosuppressive TME are depicted in Figure 2 and are described below.

### 3.1. CAF-MDSC Interactions

In PDAC patients, the number of MDSCs in the peripheral blood is related to disease progression [50]. In several studies, the involvement of CAFs in regulating MDSC infiltration and activation has been investigated. iCAF secretion of C-X-C motif chemokine ligand 12 (CXCL12) increases MDSC infiltration, while signalling activation and differentiation are promoted via additional soluble factors IL-6, VEGF, C-C chemokine (CC-motif) ligand 2 (CCL2) and macrophage colony-stimulating factor (M-CSF) [51,52]. Subsequently, MDSCs can inhibit T cells via the depletion of the amino acid L-arginine in the TME [53,54]. The immunosuppression by MDSCs in PDAC patients is dependent on the activation of STAT3, which can be activated through CAF-derived IL-6 [55]. Taken together, CAFs induce MDSC activity and in doing so suppress immune activity in the TME.

### 3.2. CAF-Macrophage Interactions

Circulating monocytes in addition to tissue-resident macrophages give rise to TAMs [56]. TAMs are commonly subclassified into M1 and M2 macrophages due to their opposite polarization states. In cancer, M1 macrophages have been identified as pro-inflammatory with anti-neoplastic effects and M2 macrophages as anti-inflammatory with pro-neoplastic effects [57]. In PDAC specifically, a high-density infiltration of M2 macrophages has been associated with a shorter OS [58]. The presence of CAFs in the TME increases signalling molecules such as IL-17, which can mediate monocyte recruitment [59,60]. In addition, convincing evidence reported in multiple studies suggests that CAFs can polarize macrophages to an M2 state. For instance, monocytes added to a 3D co-culture of pancreatic cancer cells and fibroblasts differentiated into TAMs with an M2 phenotype [61]. These TAMs could inhibit helper T cell and cytotoxic T cell activity and their proliferative capability in vitro. In line with these results, a co-culture of monocytes and CAFs showed induction of M2 polarization of macrophages by CAFs through M-CSF secretion. Similarly, CAF-dependent M2 macrophage polarization has been observed in a PDAC mouse model, in which CAFs induced M2 macrophage differentiation in an IL-33-dependent manner [62]. Finally, in vivo deletion of hypoxia-inducible factor 2 (HIF2) in CAFs reduced M2 macrophage differentiation and regulatory T cell recruitment. These findings evidence the ability of CAFs to suppress the immune system via regulating TAM polarization.

### 3.3. CAF-Neutrophil Interactions

Neutrophils are the most common systemic immune cells and play a key role in the innate immune response. In PDAC, neutrophil infiltration is increased compared to the non-cancerous pancreas and during pancreatic inflammation (pancreatitis) [63]. As for macrophages, subclasses N1 and N2 TANs have been reported. TGF-β, which also activates CAFs, is proposed to induce N2 TAN recruitment and activation. N2 TANs display immunosuppressive and pro-tumourigenic effects. As for N1 TANs, they are related to increased T cell cytotoxicity and pro-inflammatory cytokine expression [64]. In PDAC, TAN infiltration has been negatively associated with patient prognosis [63,65]. The effect of CAFs on neutrophils has been studied to a significantly lesser extent. One study proffers that CAFs can drive NETosis in neutrophils via amyloid-β expression in PDAC [66]. NETosis is the formation of neutrophil extracellular traps composed of histone-bound nuclear DNA and cytotoxic granules. Tumour-induced NETosis was linked to cancer progression, and high amyloid-β expression was associated with poor prognosis in human cancer. Inhibition of NETosis in vivo by inhibiting amyloid-β release restricted PDAC tumour growth in mice. Despite the limited number of reports thus far, CAF–neutrophil interactions seem to enhance the pro-tumourigenic effects of TANs and represent an interesting venue for further studies.

### 3.4. CAF-T Cells

T cell infiltration and T cell subset ratio differ across pancreatic tumours [67,68]. These differences in T cell infiltration have been correlated with PDAC patient outcomes. Specifically, high cytotoxic T cell or helper T cell infiltration is associated with improved survival, whereas high and regulatory T cell infiltration is associated with lower survival [48,63]. When spatial information was included in the evaluation, the proximity of cytotoxic T cells to cancer cells was associated with improved survival, suggesting that these cells play a crucial role in anti-tumour efficacy [48]. Nevertheless, many PDACs are characterized by the exclusion of cytotoxic T cells from the tumour, and low T cell density is believed to be one of the main barriers to the efficacy of immunotherapy.

Several reports have shown that CAFs can actively contribute to T cell exclusion. One study pointed towards a role for CXCL12 produced by FAP+ CAFs in T cell exclusion [48,49]. Chemical inhibition of CXCR4, the receptor of CXCL12, resulted in a rapid influx of T cells in the tumour beds. It remained unclear how CXCL12 contributes to T cell exclusion. In another study, it was demonstrated that CAFs that express FAK1 generated increased deposition of collagen and reduced cytotoxic T cell infiltration [69]. Hence, CAFs may hinder T cell infiltration via the release of CXCL12 and ECM remodelling through collagen deposition. In contrast, a relation between high cytotoxic T cell infiltration and a high stromal density was reported [70]. Possibly, parts of the ECM remodelling by CAFs that arise are beneficial, while other parts such as collagen are detrimental to anti-tumour immunity.

However, the general role of CAFs in mediating T cell suppression has now been challenged by several more recent studies, suggesting that T cell paucity cannot simply be explained by CAF-induced desmoplasia. A study investigating spatial relationships between cancer cells, cytotoxic T cells, CAFs and collagen showed that α-SMA+ CAFs or collagen-I were not enriched in areas with low cytotoxic T cells [48]. Furthermore, the shaping of the intra-tumoural T cell response may vary among the CAF subsets, resulting in immunosuppressive or immunostimulatory signals. myCAFs are mostly involved in inducing desmoplasia (1), iCAFs secrete cytokines (2) and apCAFs are involved in antigen presentation (3). As previously mentioned, decreased overall immune infiltration, yet increased regulatory T cell infiltration, was found when α-SMA+ myCAFs were selectively depleted in a transgenic mouse model of PDAC [30]. In another study to investigate regulatory T cells in PDAC in vivo, conditional cell depletion of regulatory T cells caused a loss of TGF-β expression and reduced α-SMA expression, the marker linked to the myCAF subset [71]. Hence, myCAFs and regulatory T cells seem to mutually influence each other. The studies mentioned above together suggest that myCAFs are less decisive in T cell exclusion and immunosuppression than previously assumed. As for iCAFs, the expression of IL-6 by the subset is associated with an increase in regulatory T cells [32]. CXCL12, the molecule earlier discussed that mediates T cell exclusion, is primarily associated with iCAFs [51]. In addition, an in vivo study on the signalling of CAFs and the effects on immune cells in KPC mice revealed that a knockout of IL-17A, a cytokine secreted by CAFs, reduced regulatory T cells and increased cytotoxic T cells [60]. The apCAF subset can influence the immune system via the presentation of antigens via MHC II, yet they do not have the co-stimulatory molecules necessary for immune cell activation. This causes T cell anergy and an increase in regulatory T cells [33,39]. Similarly, in colorectal cancer, it was shown that apCAFs reduced cytotoxic T cell activation, overall cytotoxicity and enhanced exhaustion [41]. Moreover, another study provided evidence that apCAFs can kill cytotoxic T cells directly in an antigen-dependent manner via the expression of PD-L2 and FASL [40].

In summary, CAF can influence T cells in various ways, impacting their differentiation, function or infiltration. Since T cells are the main effector cell of most immunotherapies, the effect CAFs have on T cells could impact their efficacy.

## 4. The Road to Immunotherapy for PDAC with CAFs as a Potential Hurdle

Immunotherapy has proven to be an effective therapeutic agent, especially in melanoma, non-small cell lung cancer and tumours with microsatellite instability [72,73]. The most abundantly studied immunotherapies are currently immune checkpoint inhibitors, which block checkpoint proteins from binding to their receptor. Other immunotherapeutic approaches investigated for PDAC include cancer vaccines, CD40 agonists, chimeric antigen receptor (CAR), T cell therapy and oncolytic viruses. The main mechanisms of these immunotherapies enhance overall immune activity or direct cytotoxic T cells to target tumour cells. As described above, interactions between CAFs and immune cells likely cause an immunosuppressive phenotype in PDAC, which could play a key part in the setbacks in immunotherapy. Below, we will discuss commonly used immunotherapeutic approaches and the current data on how CAFs might affect their efficiency.

### 4.1. Checkpoint Inhibitors

The most applied immunotherapy is immune checkpoint inhibition, which releases the ‘brakes’ put on the host T cell response against the tumour. Two of the most studied immune checkpoint molecules are cytotoxic T lymphocyte-associated protein 4 (CTLA-4) and programmed cell death protein 1 (PD-1), the latter binding to its ligand PD-L1 on cancer cells and immune cells. [74]. Checkpoint inhibitors are approved as first- or second-line therapies for an increasing number of cancer types, such as melanoma and MSI-high tumours [75]. Unfortunately, clinical trials with checkpoint inhibitors did not show clinical benefit in patients with PDAC [76,77,78]. Given the fact that the mutational burden is positively correlated with the response rates to immune checkpoint inhibitors, the low mutational burden of PDAC could explain the low efficacy [8]. Furthermore, the desmoplastic and immunosuppressive TME, and especially CAFs, could contribute to immunotherapy resistance [79].

As previously mentioned, CAFs can reduce cytotoxic T cell infiltration, and since checkpoint inhibitors target T cells, their presence is instrumental [33,80]. In PDAC patients with a low degree of desmoplasia, the response to checkpoint inhibitors was profoundly better [43], probably due to T cells being more able to infiltrate the TME. Likewise, the diphtheria toxin-mediated depletion of FAP+ CAFs that contributed to T cell exclusion in a KPC mouse model resulted in improved response to immune checkpoint inhibitors in otherwise non-responsive mice [49]. To understand ECM remodelling, a pan-cancer analysis has been executed that compares genomic and cellular alterations across cancer types between malignant and normal tissue [81]. The created ECM dysregulation signature was related to TGF-β signalling in CAFs and was linked to the failure of anti-PD-1 treatment. Accordingly, the data obtained in this study suggest that CAFs dysregulate ECM organisation via TGF-β dependent effects, thereby inhibiting T cell infiltration and thus the effects of checkpoint inhibitor therapy. In another study, a correlation between a subset of myCAFs and checkpoint inhibitor efficacy was reported in PDAC patients [82]. An elevated level of leucine-rich repeat containing 15 (LLRC15)+ myCAFs was associated with a poor response to anti-PD-L1 treatment. In murine breast cancer, a high abundance of CAFs was associated with insensitivity to anti-CTLA-4 and anti-PD-L1 therapy [83]. Furthermore, genetic depletion of a subset of myCAFs exhibiting the CAF receptor Endo180 (MRC2) led to increased cytotoxic T cell infiltration and improved checkpoint inhibitor sensitivity. In conclusion, CAFs conceivably seem to suppress the efficacy of checkpoint inhibitors via the inhibition of T cell infiltration.

### 4.2. Activating Immune Checkpoint Agonists

In addition to targeting inhibitory molecules, the activation of immunostimulatory molecules has been investigated, such as the CD40 agonists. CD40 is part of the tumour necrosis factor (TNF) receptor family and promotes T cell activation and M1 TAM polarization [84]. The first clinical trials with a CD40 agonist showed promising results in patients with non-operable PDAC [85]. Nevertheless, a phase II trial on the combination of chemotherapy, a CD40 agonist and a PD-1 inhibitor did not show increased therapeutic efficacy in the groups that received the combination with the CD40 agonist [86]. In a mouse model for colorectal cancer, the combination of a CD40 agonist with an M-CSF-1 receptor antibody increased survival compared to monotherapy [87]. This indicates that inhibiting CAF-mediated immunosuppressive M-CSF signalling might improve CD40 therapies. Nevertheless, the presence of tumour-specific T cells is still a requisite for the efficacy of immune checkpoint inhibitors or agonists.

### 4.3. Cancer Vaccines

Cancer vaccines are established anti-cancer strategies when it comes to inducing novel cancer-specific T cell responses directed against tumour-specific antigens [88]. Typical vaccines contain an immunostimulatory adjuvant and antigens conveyed in the form of mRNA, DNA, peptides or whole cells. Cancer vaccine-induced immunization starts with the uptake of the antigens by dendritic cells (DC), which sets the proliferation and differentiation of T cells in motion. In pancreatic cancer, initial results with a whole-cell vaccine seemed promising. However, the addition of the vaccine to standard care chemotherapy in a phase III trial did not improve PDAC patient survival [89]. Another approach was taken with a DC vaccine loaded with whole tumour lysate. Promising results were obtained in a mouse model for PDAC, in particular when the vaccine was combined with a CD40 agonist [90]. Data from a phase I trial on this combination in PDAC patients revealed that no adverse events occurred, and T cell activity increased [91] Although research on the effects of CAFs on cancer vaccination is limited, there are indications that CAFs can inhibit DC and T cell function, thereby hampering effective immune priming. In solid tumours, the release of TGF-β and IL-6 likely suppresses the proliferation and migratory properties of DCs, resulting in interference with tumour-directed priming of cytotoxic T cells [24]. In hepatocellular carcinomas, it was shown that activation of the IL-6-mediated STAT pathway in CAFs causes them to recruit DCs and inhibit antigen presentation [92]. In addition to directly influencing DCs, CAFs also inhibit, as described before the ultimate effector cells of immunization, cytotoxic T cells [93]. In summary, both agonistic CD40 therapies and cancer vaccines are in the early stages of clinical development and show promise, in particular in combination with other immunotherapies to overcome immunosuppression.

### 4.4. CAR T Cells

Cytotoxic T cells of patients can be genetically engineered to express receptors that specifically recognize tumour cells, thereby creating CAR T cells. CAR T cells targeting antigen CD19 were the first approved second-line treatment for lymphoma and leukaemia patients [94]. In solid tumours, CAR T cell therapy is less efficient, which could be due to the difficulty of CAR T cells to enter desmoplastic, stroma-rich PDAC tumours. The impact of CAFs on CAR T cell therapy is still speculatory. As previously described, CAFs are linked to T cell exclusion [69]. CAR T cell therapy could be considered to target CAFs. An interesting opportunity could arise from a novel CAR T cell product directed to FAP. At present, one clinical trial has been conducted in mesothelioma patients to assess the safety and feasibility of local CAR T cell therapy targeting FAP on the mesothelioma cells [95]. No treatment-related toxicities were observed, and two out of three patients were still alive after a median follow-up of 18 months. These encouraging data may open up opportunities for using FAP-targeting CAR T cells to target FAP+ CAFs in the future.

### 4.5. Oncolytic Virus Therapy

Oncolytic viruses can be used to induce oncolysis of tumour cells directly or indirectly via activation of anti-tumour immunity. Adenoviruses, herpesviruses, reoviruses and parvoviruses have been tested as oncolytic viruses in clinical trials in PDAC, but until now, oncolytic viruses as monotherapy have not shown sufficient efficiency [7]. The desmoplastic nature of the TME could play a role in reduced penetration of the virus into the tumour. The ability of oncolytic viruses to infect and kill CAFs and other stromal cells besides tumour cells could be a way to decrease the pro-tumourigenic effects of CAFs [96]. Recently, tropism by the reovirus towards human and mouse pancreatic CAFs was observed, while the same work reported that the crosstalk between CAFs and tumour cells can impact the viral infection [97]. Even though oncolytic viruses are not successful as monotherapy yet, their capacity to transform an immunosuppressive TME of solid tumours into an immune active TME with enhanced T cell influx can be used to boost the efficacy of other immunotherapeutic strategies. Proof-of-concept has been demonstrated with a combination of oncolytic reovirus and CD3-bispecific T cell engagers in a mouse model of PDAC [98].

In conclusion, it has become evident that immunotherapies in PDAC have potential but do not show desired efficiency, possibly in part due to CAFs and the immunosuppressed environment they generate.

## 5. Targeting CAFs and Their Products to Advance Immunotherapy Responses

Given the potentially important role that CAFs play in modulating responses to immunotherapy, a solution to low efficiency could be to modulate the CAF population or the CAF-mediated downstream effects in PDAC. Given the distinct subsets and their specific role in regulating immune cell trafficking and activity, this has to be considered carefully. As indicated before, the depletion of all CAFs from PDAC tumour models in mice, contrary to the expectations, generated an immunosuppressed environment [30]. These data indicate that a more selective approach is warranted. Table 1 summarizes the effects of targeting CAFs to improve the immunological effects of immunotherapies.

### 5.1. The Fight against Immunosuppression

Taking into consideration markers of the currently known CAF subsets can help identify the right targets for therapy. A decrease in iCAF differentiation or abundance and thereby iCAF-induced immunosuppression could improve immunotherapy. An inhibitor of IL-1, a promotor of iCAF differentiation, was administered together with the checkpoint inhibitor PD-1 to PDAC-bearing mice [92]. The combination therapy improved anti-PD-1 blockade sensitivity by enabling cytotoxic T cell infiltration and by attenuating tumour growth [102]. Another approach is to inhibit inflammatory cytokines secreted by iCAFs, such as IL-6. A combination of an IL-6 inhibitor and anti-PD-L1 resulted in increased cytotoxic T cell infiltration and impaired tumour progression in the PDAC KPC-Brca2 mouse model [101]. Next to directly targeting IL-6, an indirect approach has been tested by inhibition of heat shock protein 90 (hsp90) with small molecule XL-888, which decreased IL-6 secretion by CAFs [100]. Consequently, the combination of XL-888 and anti-PD-1 increased T cell infiltration and enhanced therapeutic efficacy in KPC mice [100].

IL-6 inhibition has also been applied in combination with a CD40 agonist. The treatment increased the anti-tumour effects by reduction of TGF-β, collagen, PD-L1 and PD-1 expression indicative of targeting the CAFs in the TME [110]. In addition to IL-6, iCAFs have also been targeted by inhibition of CXCL12 and its receptor C-X-C motif receptor 4 (CXCR4) [33]. In a phase II trial, the combination of a CXCR4 antagonist, anti-PD-1 and chemotherapy [118] improved cytotoxic T cell infiltration, reduced MDSCs and decreased circulating regulatory T cells [100]. These observations highlight the possibilities of inhibiting cytokines or chemokines secreted by iCAFs to reduce immunosuppression.

Recently, it was shown that iCAF-derived cytokines can activate CXCR2 on CAFs, which causes conversion into myCAFs. This phenomenon is associated with increased PDAC metastasis [121]. CXCR2 inhibition in combination with anti-PD-1 led to increased helper and cytotoxic T cell infiltration, while also reducing neutrophil and regulatory T cells [103,104]. Therefore, CAF subtype conversion might be a novel and challenging but interesting approach to improve immunotherapy.

### 5.2. Targeting CAFs Directly

Another approach would be to reduce the number of CAFs in the TME by therapeutically targeting these cells. The most abundantly studied molecule on CAFs that has been used as a target is FAP. In PDAC mouse models, CAR T cell therapy targeting FAP+ CAFs led to toxicity, while in another study, it was effective without inducing toxicity [112,113]. The fact that FAP+ is also expressed scarcely in other cells, such as bone marrow cells, must be taken into consideration. A second approach to target FAP used an adenovirus that expressed a bispecific T cell engager (biTe) that binds to FAP. Adenoviral therapy stimulated T cell activation and T cell infiltration and decreased tumour growth in PDAC-bearing mice [114]. A third approach to target FAP+ CAFs that has been evaluated is a vaccine. A vaccine against the tumour antigens survivin and FAP decreased the proportion of immunosuppressive cells and increased lymphocyte infiltration in a pancreatic cancer murine model [109]. An alternative way to target FAP+ CAFs is via antibody drug conjugates (ADCs) containing a cytotoxic payload. OMTX705, an ADC including a humanized anti-FAP antibody and the cytolytic compound TAM470, demonstrated potent tumour growth inhibition in PDX mouse models with various solid tumours, including PDAC [117]. Mechanistically, OMTX705 resulted in stromal cell depletion in PDAC models, as indicated by reduced α-SMA and Col11a1 staining and increased cytotoxic T cell infiltration. Lastly, a FAP-CD40 antibody that exclusively initiated anti-tumour immune activity in the presence of FAP reduced tumour growth in PDAC-bearing mice [111]. Besides FAP, a marker of interest expressed by CAFs could be peptidyl-prolyl isomerase NIMA-interacting 1 (PIN1). Upregulated PIN1 in CAFs leads to desmoplasia and thereby poor cytotoxic T cell infiltration. Inhibitors of PIN1 improved anti-PD-1 response in KPC mice and increases survival [99].

In conclusion, there are attractive targets in PDAC that can be used to relatively, specifically inhibit CAFs or as a docking site for therapeutic antibodies directed towards the TME. Further studies in PDAC patients remain warranted to elucidate potential mechanisms and efficiency.

### 5.3. Targeting CAFs from Multiple Angles

Besides immunosuppressive influences in the TME, CAFs are known for remodelling the ECM and inducing desmoplasia. Consequently, approaches have been evaluated to target both the immunosuppressive and desmoplastic effects of CAFs. Hyaluronan is a component of the ECM and is associated with α-SMA+ CAFs [122]. Hyaluronan inhibition could pave the way for immunotherapy by reducing desmoplasia and allowing for anti-tumour immunity. Administration of a hyaluronan inhibitor and whole cell PDAC vaccine, GVAX, in vivo inhibited immunosuppression via the reduction of CXCL12/CXR4 signalling, decreased desmoplasia and conferred a survival advantage in comparison with single-agent therapies [108]. Along similar lines, oncolytic adenovirus VCN-01, which includes hyaluronidase to improve virus intra-tumoural spread and anti-tumour immunity, was administered in combination with chemotherapy to PDAC patients in a phase I trial [120]. No dose-limiting toxicity was observed, and an overall response rate of 50% was detected. These results support the premise that reducing desmoplasia improves anti-tumour immunity.

TGF-β is involved in myCAF promotion and promoting desmoplasia while also having immunosuppressive effects. Therefore, TGF-β inhibition is an attractive approach to improve the TME of PDAC on multiple levels, including the reduction of desmoplasia and immunosuppression. A number of studies have used TGF-β inhibition in PDAC mouse models. In KPC mice, the combination of PD-L1 and a TGF-β receptor small molecule inhibitor delayed tumour growth relative to untreated mice but did not increase anti-tumour immune response [116]. In contrast, the combination therapy elicited an immune response in colorectal cancer-bearing mice in this study. As an alternative tactic, TGF-β-derived peptide vaccination was tested, which enhanced cytotoxic T cell infiltration, M1 TAM polarization and reduced myCAF abundance in PDAC-bearing mice [107]. Another study intended to enhance the efficacy of immunotherapy via the addition of a TGF-β-blocking antibody to a reovirus and via CD3-bispecific antibody combination therapy [123]. The TGF-β-blocking antibody antagonized the therapeutic efficacy of the two other treatments. The TGF-β blockade has been investigated in metastatic PDAC patients in a phase I study as well. Treatment with a TGF-β inhibitor and anti-PD-L1 did not give rise to serious adverse events, but the OS remained low, which might be related to the advanced tumour stage [119]. Accordingly, different responses are encountered after TGF-β inhibition, possibly due to the pleiotropic role of TGF-β in the TME.

Besides TGF-β, vitamin D could be targeted to reprogram the TME. Three-dimensional in vitro models showed that vitamin D decreases CAF proliferation and migration yet upregulates PD-L1 [105]. In a PDAC mouse model, activation of the vitamin D receptor decreased desmoplasia in vivo [124]. Although, vitamin D can initiate Th2 and inhibit Th1 immune responses, which further enables the immunosuppressive state of the TME [125]. Thus, vitamin D therapy could reduce desmoplasia promoted by CAFs but restrict the activity of anti-tumourigenic T cells via promoting immune suppression.

This evidence emphasizes the challenging interconnections that signalling cascades have. One pathway could both increase pro-tumourigenic immunosuppression and decrease pro-tumourigenic desmoplasia. Consequently, the network of CAFs must first be understood to know its entire effect on immunotherapy.

## 6. Concluding Remarks

To improve PDAC treatment options and survival, it is essential to understand the intricacies of the TME, including immune cell and CAF interactions. The current literature provides a solid foundation for the notion that CAFs are instrumental in evoking immunotherapy resistance, but the underlying mechanisms are insufficiently explored. The major question remains: are CAFs the good or the bad guys in PDAC? Pathways associated with iCAFs are predominantly, but not exclusively, deemed as pro-tumourigenic. CAFs are involved in desmoplasia, but depletion of myCAFs can lead to tumour progression. This poses a major hurdle for current and future efforts. Mapping CAF subsets and exploring their function and plasticity are key before exploring or excluding therapeutic opportunities. Thus far, the combination of CAF-targeting therapies with immunotherapies shows promising effects in murine models and some clinical trials. These studies pave the way for the future of PDAC treatment.

## Figures and Tables

**Figure 1 ijms-24-08707-f001:**
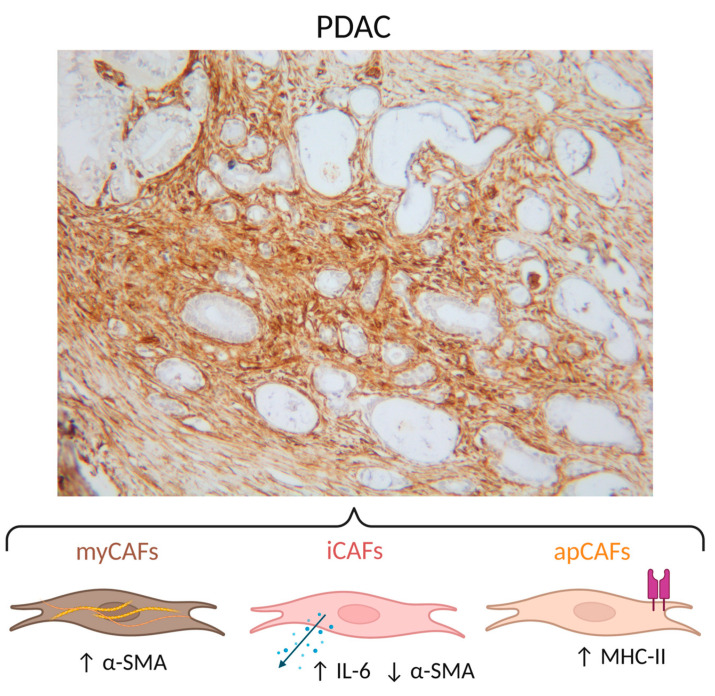
The most common subclassification of CAFs in PDAC consists of myCAFs (α-SMA+), iCAFs (IL-6 secretion, α-SMA−) and apCAFs (MHC-II). PDAC: pancreatic ductal adenocarcinoma; CAF: cancer-associated fibroblast; myCAF: myofibroblast-type CAF; iCAF: inflammatory CAF; apCAF: antigen-presenting CAF; α-SMA: α-smooth muscle actin; IL-6: interleukin 6; MHC-II: major histocompatibility complex class II. Histology picture: human PDAC sample stained for vimentin. Created with BioRender.com (www.biorender.com, accessed on 3 April 2023).

**Figure 2 ijms-24-08707-f002:**
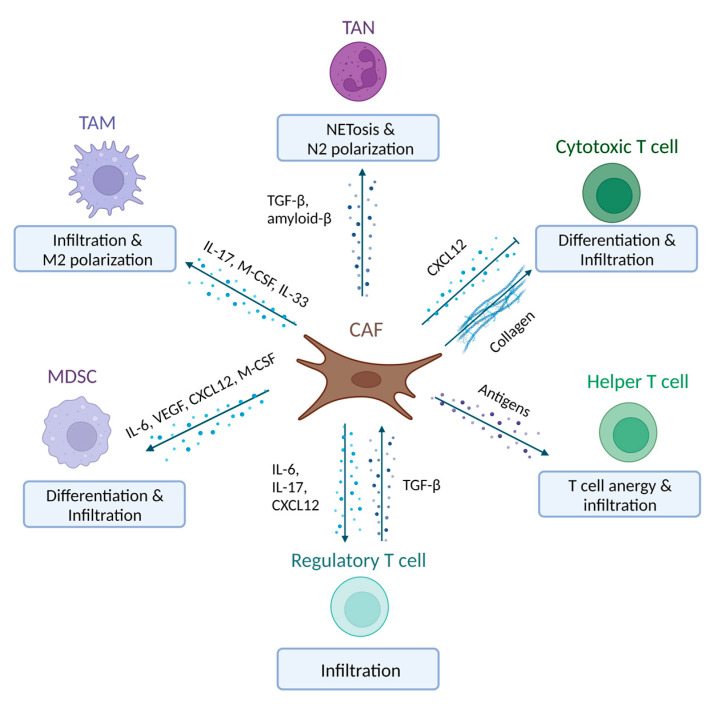
Schematic representation of the observed CAF–immune cell interactions in PDAC. CAFs alter the function, differentiation and infiltration of various myeloid cells, such as TANs, TAMs and MDSCs, as well as T cells. CAF: cancer-associated fibroblast; PDAC: pancreatic ductal adenocarcinoma; TAN: tumour-associated neutrophil; TAM: tumour-associated macrophage; MDSC: myeloid-derived suppressor cell; TME: tumour microenvironment; TGF-β: transforming growth factor β; IL-17: interleukin 17; M-CSF: macrophage colony-stimulating factor; VEGF: vascular endothelial growth factor; CXCL12: C-X-C motif chemokine ligand 12. Created with BioRender.com (www.biorender.com, accessed on 3 April 2023).

**Table 1 ijms-24-08707-t001:** Overview of studied CAF targets in the PDAC TME with the aim to improve immunotherapy. Results from preclinical models as well as clinical trials are summarized. Effects on CAFs, immunity, tumour growth and survival are separately indicated. ↓: a decrease; ↑: an increase; -: no observed difference; n.d.: no data; CAF: cancer-associated fibroblast; PIN1: peptidyl-prolyl isomerase NIMA-interacting 1; PD-1: programmed cell death protein 1; KPC: Kras^LSL-G12D^, Trp53^LSL-R172H^, Pdx1-cre; hsp90: heat shock protein 90; IL-6: interleukin 6; PD-L1: programmed cell death protein ligand 1 (PD-L1); KPC-brca2: Kras^LSL-G12D^, Trp53^LSL-R270H^, Pdx1-cre, Brca2^F/F^; CXCR2: chemokine receptor 2; FAP: fibroblast activation protein; TGF-β: transforming growth factor β; CAR: chimeric antigen receptor; NSG: NOD/scid/IL2rg−/−; ADC: antibody drug conjugate.

Immunotherapy	Target	Design	CAFs	Immunity	Tumour Growth	Survival	Reference
Preclinical studies							
Checkpoint inhibitor	PIN1 + PD-1	*KPC*	↓	↑	↓	↑	[99]
	hsp90 + PD-1	*KPC*	↓	↑	↓	n.d.	[100]
	IL-6 + PD-L1	*KPC-Brca2*	n.d.	↑	↓	↑	[101]
	IL-1 + PD-1	*KPC*	n.d.	↑	↓	↑	[102]
	CXCR2 + PD-1	*KPC*	n.d.	↑	↓	↑	[103,104]
	Vitamin D analogue + PD-L1	*2D + 3D culture*	↓	↓	-	-	[105,106]
	FAP + PD-L1	*KPC*	↓	↑	↓	n.d.	[49]
Vaccine	TGF-β	*C57BL/6*	↓	↑	↓	n.d.	[107]
	Hyaluronan + GVAX	*C57BL/6*	↓	↑	↓	↑	[108]
	FAP + survivin	*C57BL/6*	↓	↑	↓	↑	[109]
CD40 agonist	IL-6 + CD40 agonist	*C57BL/6*	n.d.	↑	↓	↑	[110]
	FAP + CD40 agonist	*KPC-huCEA*	↓	↑	↓	n.d.	[111]
CAR T cell	FAP	*C57BL/6*	n.d.	↑	n.d.	n.d.	[112]
	FAP	*KPC*	↓	↑	↓	↑	[113]
Oncolytic virus	FAP	*NSG*	↓	↑	↓	↑	[114]
	CAF + tumour cells	*BALB/c-nu/nu*	n.d.	↑	↓	↑	[115]
	TGF-β	*KPC*	n.d.	-	-	-	[116]
	TGF-β + CD3 + tumour cells	*KPC*	↓	-	-	-	[98]
ADC	FAP + tumour cells	*Foxn1 nu/nu*	↓	↑	↓	↑	[117]
Clinical trials							
Checkpoint inhibitor	CXCR4 + PD-1	Phase II	n.d.	↑	↓	↑	[118]
	TGF-β + PD-L1	Phase I	n.d.	n.d.	n.d.	n.d.	[119]
Oncolytic virus	Hyaluronan + tumour cells	Phase I	n.d.	↑	↓	↑	[120]

## Data Availability

Not applicable.

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
