# Peer review of "The Tango between Cancer-Associated Fibroblasts (CAFs) and Immune Cells in Affecting Immunotherapy Efficacy in Pancreatic Cancer"

_ijms, 2023, doi:10.3390/ijms24108707_

Round 1
Reviewer 1 Report
This review focuses on exploring the suppressive effects of CAF in cancer and discusses potential strategies for targeting CAF in cancer immunotherapy, including checkpoint inhibitor and CAR-T cell therapies. The review provides a comprehensive insight into the subject matter, however, to enhance the manuscript's soundness, the following comments are suggested:
1. It would be helpful if the authors briefly introduce the composition and functions of CAFs in other cancer types, in addition to pancreatic ductal adenocarcinoma (PDAC), and highlight the differences between hot and cold tumors.
2. Table 1 could be improved by separating clinical trials and preclinical studies into two separate sections, or by generating two distinct tables.
3. The authors may want to consider discussing the recently developed FAP ADC.
4. In discussing the advantages of targeting FAP in CAF, the authors could also include a discussion of CAR-NK and CAR-iNKT cells.
No
Reviewer 2 Report
The interplay between CAFs and the immuine system is a very important and timely topic for a review in light of the increasing importance of immunotherapies across multiple cancers. The current review provides a good overview of the current satus of research for pancreatic adenocarcinoma.
It is well organized and well written and an interesting read.
Minor point: In lines 397 to 400, data from a CART cell therapy targeting FAP in mesothelioma are discussed. It might be woth mentioning here that FAP is typically not only expressed on CAFs but also on mesothelioma cells.
